# Prospective Representation Learning for Non-Exemplar Class-Incremental Learning

**Wuxuan Shi**[1], **Mang Ye**[1,2*]

[1]School of Computer Science, Wuhan University, Wuhan, China
[2] Taikang Center for Life and Medical Sciences, Wuhan University, Wuhan, China
{wuxuanshi, yemang}@whu.edu.cn
https://github.com/ShiWuxuan/NeurIPS2024-PRL

## Abstract

Non-exemplar class-incremental learning (NECIL) is a challenging task that requires recognizing both old and new classes without retaining any old class samples. Current works mainly deal with the conflicts between old and new classes retrospectively as a new task comes in. However, the lack of old task data makes balancing old and new classes difficult. Instead, we propose a Prospective Representation Learning (PRL) approach to prepare the model for handling conflicts in advance. In the base phase, we squeeze the embedding distribution of the current classes to reserve space for forward compatibility with future classes. In the incremental phase, we make the new class features away from the saved prototypes of old classes in a latent space while aligning the current embedding space with the latent space when updating the model. Thereby, the new class features are clustered in the reserved space to minimize the shock of the new classes on the former classes. Our approach can help existing NECIL baselines to balance old and new classes in a plug-and-play manner. Extensive experiments on several benchmarks demonstrate that our approach outperforms the state-of-the-art methods.

## 1 Introduction

In recent years, deep neural networks (DNNs) have achieved great success in static scenarios. Research attention is increasingly turning to extending the learning capability of DNNs to open and dynamic environments. An important aspect is to enable the network to accumulate knowledge from new tasks as the input stream is updated (*i.e.*, incremental learning [1; 2; 3]).

Whenever a new task arrives, it is costly to retrain the model with current and old data. Not to mention that the old data is not fully available. A typical alternative is to fine-tune the network with new data directly. However, this can lead to drastic performance degradation on previously learned tasks, a phenomenon known as catastrophic forgetting [4; 5]. While storing exemplars of each class is a simple approach to mitigate forgetting, it relies on the quality of saved exemplars and faces challenges in storage and privacy, especially for sensitive domains such as medical imaging. Hence, this paper focuses on coping with catastrophic forgetting during incremental learning without storing any old samples, which is called non-exemplar class-incremental learning (NECIL) [6; 7].

In NECIL, a serious challenge is to discriminate between old and new classes without access to old data. Most methods usually start considering handling conflicts between old and new classes only when new tasks arrive. While some methods use stored prototypes to model the distribution of old classes [8; 6; 9; 10], others extend the network structure to accommodate new classes [7; 11]. However, in the base phase (*i.e.*, training on the first task), traditional training allows different classes

---

*Corresponding Author: Mang Ye

38th Conference on Neural Information Processing Systems (NeurIPS 2024).

to divide up all the embedding space, causing trouble for subsequent conflict resolution. As shown in Fig. 1, in the incremental phase (*i.e.*, training on tasks after the first one), with the influx of new classes, there are overlaps of the old and new classes in the embedding space that are difficult to discriminate. Moreover, due to the unavailability of old class samples, handling this conflict with only new task data is intractable. Instead, we suggest addressing this issue by learning prospectively at the feature level, which requires a two-pronged effort in both the base and incremental phases.

Firstly, the model should make room in advance for the incoming classes in the future. Thus, the space of past classes does not need to be drastically squeezed when expanding new classes. To this end, during the base phase, we construct a preemptive embedding squeezing constraint to enforce intra-class concentration and inter-class reserved separation. Specifically, we push instances from the same class closer together and instances from different classes farther apart in a mini-batch. It allows more space to be reserved in the initial embedding space, thus making the model ready for future classes.

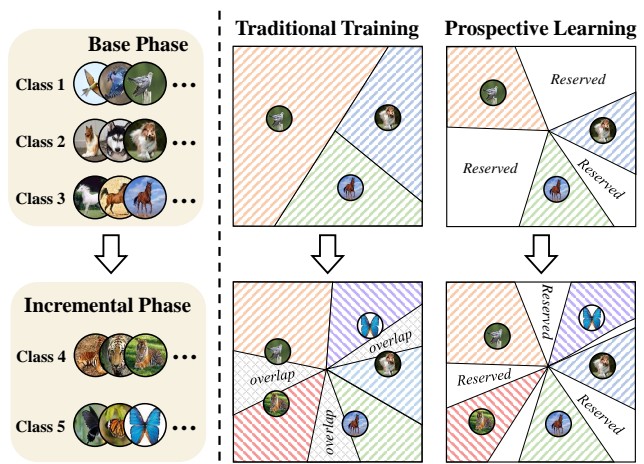

Secondly, the model should minimize the shock and impact of the new classes on the past classes, *i.e.*, embed the new classes into the reserved space as much as possible. However, achieving the desired embedding of new classes when the old class data is fully unavailable is difficult. Inspired

Figure 1: The traditional training paradigm in NECIL considers conflicts between old and new classes only when new classes arrive and is prone to overlap. We suggest prospective learning to reduce conflicts: (1) reserve space for unknown classes; (2) make the newly coming class embedded in the reserved space.

by previous works [6; 10], we try to accomplish this using prototypes (typically the class mean in the deep feature space) saved for each old class. During the incremental phase, we propose a prototype-guided representation update mechanism. Concretely, we use the network learned from previous tasks to extract features from new task samples and project these features and the saved prototypes into a latent space. In the latent space, the new class features are pushed away from the region hosting the old class and embedded as much as possible in the reserved space with the help of prototypes. We guide the update of the current model representation through the latent space to reduce the shock of the new classes on the former classes.

In summary, combining the above two ideas, our Prospective Representation Learning (PRL) scheme makes the following main contributions:

- We impose a preemptive embedding squeezing constraint to reserve space for future classes by reinforcing intra-class concentration and inter-class reserved separation.

- We propose a prototype-guided representation update strategy that utilizes the saved prototypes to reduce the impact of expanding new classes on old ones.

- Extensive experiments on four benchmarks suggest the superior performance of our approach over the state-of-the-art. We also provide a detailed analys of our method.

## 2 Related Work

### 2.1 Class-Incremental Learning

Mainstream CIL methods can be roughly divided into three categories: rehearsal-based methods, regularization-based methods, and structure-based methods.

**Rehearsal-based methods** store a portion of seen data in a fixed-size memory buffer and replay it as new data arrives. Based on the stored data, some works use knowledge distillation techniques to protect existing knowledge [12; 13; 14; 15], while others regularize the gradient to make more efficient use of the stored samples [16; 17; 18]. Additionally, several works design new strategies for memory management instead of simple random sampling. [19; 20; 21; 22]. Although rehearsal-based methods effectively mitigate catastrophic forgetting, they are encumbered by privacy concerns and become impractical under stringent storage constraints.

**Regularization-based methods** estimate the importance of different parameters for past tasks and then limit the updating of these important parameters when learning new tasks [23; 24; 25; 26]. In incremental learning, the storage of importance weights becomes essential. However, these methods are encumbered by constraints on model parameters, consequently impeding knowledge transfer and leading to suboptimal performance, particularly in long-sequence task streams.

**Structure-based methods** accommodate knowledge from new tasks by dynamically modifying the network structure. Some works extend the network by assigning new parameters of different forms to new tasks [27; 28; 29; 30]. While this approach adeptly manages extended task sequences and sustains the performance of established classes, the linear growth of network parameters with the number of tasks and the necessity for reasoning across multiple forward propagations pose significant challenges. Parameter fusion [31] and selecting partial parameters for expansion [32] mitigates this problem to some extent. An alternative is to mask part of the parameters that are highly correlated with the previous task at the parameter level or unit level [33; 34; 35; 36]. Their performance is limited by the backbone obtained on the first task.

## 2.2 Non-Exemplar Class-Incremental Learning

Recently, some works begin to focus on NECIL [8; 37; 38; 39; 40; 41; 42; 43; 44; 45; 46], due to privacy and memory concerns [47; 48; 49], where the algorithms have no access to any past data. Li *et al.* [50] combine knowledge distillation with fine-tuning in a first attempt at incremental learning without a memory buffer. Zhu *et al.* [38] propose class augmentation and semantic augmentation to address the representation bias and classifier bias caused by the lack of old task data. Yin *et al.* [37] use model inversion technology to generate samples from previous tasks to alleviate forgetting. Based on [37], Gao *et al.* [39] introduce relation-guided knowledge distillation to address the distributional gap between generated data and real data.

Zhu *et al.* [6] combat catastrophic forgetting for the first time by preserving prototypes and augmenting them. Yu *et al.* [8] address the problem of prototype outdating in the current representation space by estimating the semantic drift of past tasks and compensating for it. Furthermore, Toldo *et al.* [9] subdivide the drift into feature drift and semantic drift and compensate for both, thereby achieving better results. Shi *et al.* [10] inject information about the current feature distribution into the prototype to model the distribution of past tasks. Wang *et al.* [51] improve the prototype augmentation method based on density to make the model more focused on features of the old class with low density. Malepathirana *et al.* [52] use the domain information obtained from topological relations to optimize prototype augmentation to reduce inter-class overlap. However, previous works deal with conflicts between old and new classes only after the new data arrives and lack prospective consideration.

## 2.3 Embedding Space Regularization

Embedding space regularization has been extensively studied in literature [53; 54; 55; 56; 57]. Chaudhry *et al.* [53] first propose learning tasks in different (low-rank) embedding subspaces that are kept orthogonal to each other. They learn an isometric mapping by formulating network training as an optimization problem on the Stiefel manifold. Another idea is to implement orthogonality in the gradient space. Saha *et al.* [58] analyze network representations after learning each task with Singular Value Decomposition (SVD) to find the basis of the subspaces and store them in memory. Moreover, several methods promote forward compatibility through regularization in the initial phase. Zhou *et al.* [59] assign virtual prototypes to compress embeddings of known classes and reserve space for new classes. Shi *et al.* [60] encourage initial CIL learners to generate representations that are similar to those of models trained jointly on all classes. Compared to previous works, we target CIL in exemplar-free scenarios (NECIL). We consider how to resolve conflicts between new and old classes during the incremental phases, in addition to reserving space in the initial phase.

# 3 Methodology

## 3.1 Problem Statement

The goal of NECIL is to continually train a unified model over a series of tasks to recognize all classes learned so far. The data stream can be defined as $D = \{D_0, D_1, \ldots D_T\}$, where $T$ is the number of incremental phases. At any phase $i$, the training set $D_i$ consists of the sample set $X_i (0 \leq i \leq T)$ and the label set $Y_i$. In particular, the classes of all phases are disjoint, *i.e.*, $Y_i \cap Y_j = \varnothing, \forall i \neq j$. It is notable that only $D_t$ is available at current phase $t$. There are no old training sets (*i.e.*, $D_{0:t-1}$) to access or save in memory. To facilitate analysis, we represent the model with two components: a feature extractor $\mathcal{F}$ with parameters $\theta$ and a unified classifier $\mathcal{G}$ with parameters $\varphi$. For a comprehensive evaluation of the model, the test set at phase $t$ includes classes from all the seen label sets $Y_0 \cup Y_1 \cup \ldots \cup Y_t$. At the time of testing, the model does not have access to the task ID, *i.e.*, it does not know from which task the test sample come.

## 3.2 Baseline

We adapt the paradigm of existing NECIL works [6; 7; 11; 10] as our baseline, which primarily uses knowledge distillation and prototype rehearsal. Specifically, at the base phase (*i.e.*, $t = 0$), the classification model is optimized under full supervision:

$$\operatorname*{argmin}_{\theta_t, \varphi_t} \mathcal{L}_t = \mathcal{L}_{ce}(\theta_t, \varphi_t; D_t) = - \operatorname*{\mathbb{E}}_{(x,y) \sim D_t} [y \cdot \log (\mathcal{G}_{\varphi_t}(\mathcal{F}_{\theta_t}(x)))], \tag{1}$$

where $\mathcal{L}_t$ represents the overall loss function, $\mathcal{L}_{ce}$ is the cross-entropy loss.

At the incremental phase (*i.e.*, $t > 0$), standard fully supervised training seeks to minimize the following objective:

$$\mathcal{L}_t = \mathcal{L}_{ce}(\theta_t, \varphi_t; D_{0:t-1}) + \mathcal{L}_{ce}(\theta_t, \varphi_t; D_t). \tag{2}$$

However, this is especially challenging since previous training sets $D_{0:t-1}$ are assumed to be unavailable in the NECIL setting. The absence of the first term in eq. (2) leads to a bias in favor of current classes in the feature extractor $\mathcal{F}_{\theta_t}$ and the classifier $\mathcal{G}_{\varphi_t}$. To address this problem, existing methods [38; 6; 10] adopt knowledge distillation and prototype rehearsal to cope with the bias. Specifically, they take the frozen feature extractor $\mathcal{F}_{\theta_{t-1}}$ from the previous phase $t - 1$ as a teacher and the current one $\mathcal{F}_{\theta_t}$ as a student. A distillation term is introduced to encourage the model to mimic the previous representation:

$$\mathcal{L}_{kd}(\theta_t; \theta_{t-1}, D_t) = \sum_{x \in X_i} \|\mathcal{F}_{\theta_t}(x) - \mathcal{F}_{\theta_{t-1}}(x)\|_2, \tag{3}$$

where $\| \cdot \|_2$ denotes Euclidean distance. Knowledge distillation helps maintain existing knowledge in $\mathcal{F}_{\theta_{t-1}}$, thus mitigating bias in the current feature extractor.

For the bias in the classifier, we use class-representative prototypes [6] to balance the optimization. Specifically, after the training of $t - 1$ phase, we compute a prototype $\boldsymbol{p}^c$ for each class $c$:

$$\boldsymbol{p}^c = \operatorname*{\mathbb{E}}_{(x,y) \sim D_{t-1}} [\mathcal{F}_{\theta_{t-1}}(x) \mid y = c]. \tag{4}$$

All prototypes of learned classes $\boldsymbol{P}_{0:t-1} = \{\boldsymbol{p}^c, c\}_{c \in Y_{0:t-1}}$ are stored in memory. In each training iteration of current phase $t$, existing works [6; 38; 10] augment the memorized prototypes $\boldsymbol{P}_{0:t-1}$ to $\tilde{\boldsymbol{P}}_{0:t-1}$ and train the classifier jointly with the current data $D_t$. In particular, the prototypes are involved in the standard classification optimization with the following objective:

$$\mathcal{L}_{pro}(\varphi_t; \tilde{\boldsymbol{P}}_{0:t-1}) = - \operatorname*{\mathbb{E}}_{(\tilde{\boldsymbol{p}}^c, c) \sim \tilde{\boldsymbol{P}}_{0:t-1}} [c \cdot \log (\mathcal{G}_{\varphi_t}(\tilde{\boldsymbol{p}}^c))]. \tag{5}$$

Compared to exemplar rehearsal, prototype rehearsal is more memory efficient and privacy secure. In conclusion, the overall loss function of the baseline can be expressed as:

$$\mathcal{L}_t = \mathcal{L}_{ce}(\theta_t, \varphi_t; D_t) + \alpha_1 \mathcal{L}_{kd}(\theta_t; \theta_{t-1}, D_t) + \alpha_2 \mathcal{L}_{pro}(\varphi_t; \tilde{\boldsymbol{P}}_{0:t-1}), \tag{6}$$

where $\alpha_1$ and $\alpha_2$ are the weights of the distillation loss and prototype loss, respectively. The specific implementation of prototype augmentation is not our focus. In this paper, we implement our approach based on the pipeline in PRAKA [10]. Our method can be incorporated with different augmentations and plugged into other baselines, such as PASS [6] and IL2A [38].

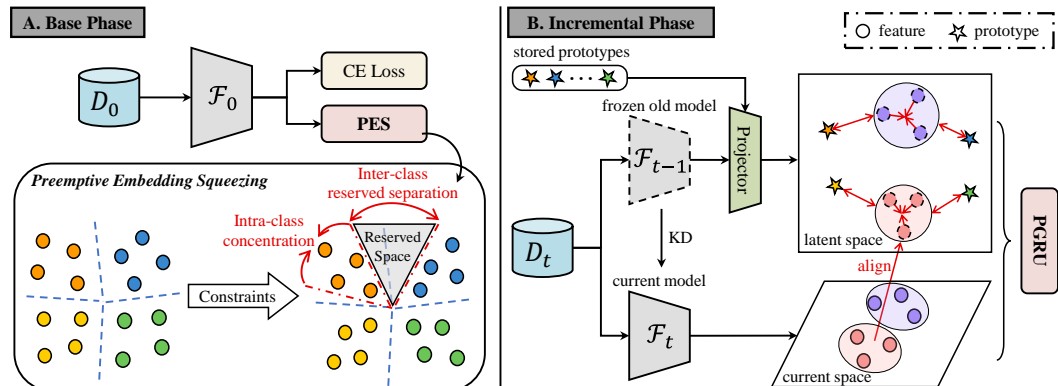

Figure 2: Overview of our Prospective Representation Learning (PRL) for NECIL. (A) During the base phase, we impose a preemptive embedding squeezing (PES) constraint to squeeze the space of the current class in preparation for accepting future new classes. (B) During the incremental phase, a prototype-guided representation update (PGRU) strategy is proposed to keep new class features away from old class prototypes in the latent space, which guides the update of the current model to mitigate the confusion of new classes with old classes.

## 3.3 Prospective Representation Learning

An overview of our Prospective Representation Learning scheme is shown in Fig. 2. It consists of a preemptive embedding squeezing constraint in the base phase and a prototype-guided representation update strategy in the incremental phase. The specific implementation of the two components is described in the following.

**Preemptive Embedding Squeezing.** In the base phase ($t = 0$), a common training paradigm of NECIL is to optimize the empirical loss over the training set $D_t$ as eq. (1). Without consideration of the future incremental learning, it overspreads the embedding space. As new classes come in, the embedding of old classes needs to be squeezed to make room for new ones. However, striking a balance in this process is challenging, especially without the old data. Therefore, we would like to be proactive and reserve space for future classes by squeezing the embedding of current classes in the base phase. Specifically, we impose a preemptive embedding squeezing (PES) constraint to cluster features of the same class and make features of different classes separate from each other. To reduce complexity, the PES loss is computed over a mini-batch data $B = \{x^i, y^i\}_{i=1}^n \in D_t$, which can be formulated as:

$$s = \sum_{\substack{\forall x^i, x^j \in B \\ y_i = y_j}} \langle \mathcal{F}_{\theta_t}(x^i), \mathcal{F}_{\theta_t}(x^j) \rangle, \tag{7}$$

$$d = \sum_{\substack{\forall x^i, x^k \in B \\ y_i \neq y_k}} \langle \mathcal{F}_{\theta_t}(x^i), \mathcal{F}_{\theta_t}(x^k) \rangle, \tag{8}$$

$$\mathcal{L}_{PES}(\theta_t; D_t) = (1 - s) + \lambda * (1 + d), \tag{9}$$

where $n$ is the batch size, $\langle \cdot, \cdot \rangle$ denotes the cosine similarity operator. As $\mathcal{L}_{PES}$ is minimized, the first term $(1 - s)$ facilitates intra-class concentration, and the second term $(1 + d)$ aims to reinforce inter-class reserved separation, as shown in Fig. 2 (A). Since $s, d \in [-1, 1)$, both terms are greater than zero. The hyper-parameter $\lambda$ controls the priority ratio of intra-class constraints and inter-class constraints. Since our PES is implemented in a vectorized manner on the mini-batch, it does not incur excessive computational burden.

With the preemptive embedding squeezing constraint, the optimization objective for the base phase training in eq. (1) can be rewritten as:

$$\mathcal{L}_t = \mathcal{L}_{ce}(\theta_t, \varphi_t; D_t) + \gamma * \mathcal{L}_{PES}(\theta_t; D_t). \tag{10}$$

where $\gamma$ is a hyperparameter controlling the weights of loss.

---

**Algorithm 1** Proposed Method

---

    **input:** Data streams $D$, Model $\{\mathcal{F}_\theta, \mathcal{G}_\varphi\}$, Factors $\lambda$ and $\gamma$, Projector $\mathcal{P}_{\phi_t}$
1: **for all** phases $t \in \{0, 1, .., T\}$ **do**
2:     Get training set $D_t$
3:     **for** minibatch $B = \{x^i, y^i\}_{i=1}^n \in \mathcal{D}_t$ **do**
4:         **if** $t = 0$ **then**
5:           Compute $\mathcal{L}_t = \mathcal{L}_{ce} + \gamma * \mathcal{L}_{PES}$
6:           Update model $\{\mathcal{F}_{\theta_t}, \mathcal{G}_{\varphi_t}\}$
7:         **else**
8:           Get prototypes set $\boldsymbol{P}_{0:t-1}$
9:           Compute $\mathcal{L}_t = \mathcal{L}_{ce} + \alpha_1 \mathcal{L}_{kd} + \alpha_2 \mathcal{L}_{pro} + \alpha_3 \mathcal{L}_{PGRU}$
10:          Update model $\{\mathcal{F}_{\theta_t}, \mathcal{G}_{\varphi_t}\}$ and projector $\mathcal{P}_{\phi_t}$
11:         **end if**
12:     **end for**
13:     Compute $\boldsymbol{p}^c = \underset{(x,y)\sim D_t}{\mathbb{E}}[\mathcal{F}_{\theta_t}(x) \mid y = c]$
14:     Update prototypes set $\boldsymbol{P}_{0:t-1}$
15: **end for**
16: **return** Model $\{\mathcal{F}_{\theta_t}, \mathcal{G}_{\varphi_t}\}$

---

**Prototype-Guided Representation Update.** In the incremental phase ($t > 0$), we would like to embed the new class into the previously reserved space. The plain idea is to keep the new classes well clustered and distanced from the old ones. To this end, we propose a prototype-guided representation update (PGRU) strategy, as shown in Fig. 2 (B), which employs prototypes as proxies for past classes to guide the embedding of new classes into the appropriate space. However, it is not practical to establish a relationship directly between the saved prototypes and the new class features extracted by the current model due to the continual updating of the current embedding space. To mitigate the mismatch between the old class prototype and the new class features, on the one hand, we use the frozen model from the previous phase $t - 1$ to extract the new class features, which has been implemented in the baseline as shown in eq. (3); on the other hand, the new classes features and the saved prototypes are projected into a unified latent space. Then, we construct orthogonal structures between the new class features and the old class prototypes in the latent space:

$$\mathcal{L}_{ort} = \sum_{\substack{\forall x^i \in B \\ \forall \boldsymbol{p}^c \in \boldsymbol{P}_{0:t-1}}} |\langle \mathcal{P}_{\phi_t}(\mathcal{F}_{\theta_{t-1}}(x^i)), \mathcal{P}_{\phi_t}(\boldsymbol{p}^c)\rangle|, \tag{11}$$

where $|\cdot|$ denotes the absolute value operator, $\mathcal{P}$ is a projector with parameters $\phi$. Similarly, $\mathcal{L}_{ort}$ is also implemented in the mini-batch to reduce computational costs. Inspired by [61], we use a simple undercomplete autoencoder as the projector. It consists of a linear layer followed by ReLU activation that maps the features to a low-dimensional subspace and another linear layer followed by sigmoid activation that maps the features back to high dimensions. When minimizing $\mathcal{L}_{ort}$, it will promote orthogonality between the new class features and the old class prototypes. By the above operations, we would like to allow the new class of features to be embedded in appropriate positions and to keep clustering in the latent space.

Our ultimate goal is to guide the update of the current model. Hence, we align the current embedding space with the latent space as:

$$\mathcal{L}_{align} = \sum_{x \in X_i} \mathcal{L}_{MSE}(\mathcal{P}_{\phi_t}(\mathcal{F}_{\theta_{t-1}}(x^i)), \mathcal{F}_{\theta_t}(x^i)), \tag{12}$$

where $\mathcal{L}_{MSE}$ is mean squared error (MSE) loss. In summary, the PGRU loss can be defined as:

$$\mathcal{L}_{PGRU} = \mathcal{L}_{ort}(\phi_t; D_t, \boldsymbol{P}_{0:t-1}) + \mathcal{L}_{align}(\theta_t, \phi_t; D_t). \tag{13}$$

In the incremental phase, the optimization objective in eq. (6) can be rewritten as:

$$\begin{aligned} \mathcal{L}_t =& \mathcal{L}_{ce}(\theta_t, \varphi_t; D_t) + \alpha_1 \mathcal{L}_{kd}(\theta_t; \theta_{t-1}, D_t) + \\ & \alpha_2 \mathcal{L}_{pro}(\varphi_t; \tilde{\boldsymbol{P}}_{0:t-1}) + \alpha_3 \mathcal{L}_{PGRU}(\theta_t, \phi_t; D_t, \boldsymbol{P}_{0:t-1}). \end{aligned} \tag{14}$$

The main procedure is summarized in algorithm 1.

# 4 Experiment

## 4.1 Experimental Setting

**Dataset.** We conduct comprehensive experiments on four public datasets: CIFAR-100 [62], TinyImageNet [63], ImageNet-Subset and ImageNet-1K [64]. CIFAR-100 consists of 100 classes, where each class contains 500 training images and 100 testing images with size 32×32. TinyImageNet has 200 classes in total, and the image size is 64×64. Each class in TinyImageNet contains 500 training images and 50 testing images. ImageNet-1K is a large-scale dataset comprising about 1.28 million images for training and 50,000 for validation with 500 images per class. ImageNet-Subset is a 100-class subset randomly chosen (random seed 1993) from the original ImageNet-1K. The image size of ImageNet-1K is much larger than the other two datasets, which poses a test of sensitivity to large-scale data.

**Protocol.** Following the setting in [6; 7; 10], we divide around half the classes for the base phase, and the rest are divided equally into all the incremental phases. *For CIFAR-100 and ImageNet-Subset*: 1) 50 classes for base phase and 5 incremental phases of 10 classes; 2) 50 classes for base phase and 10 incremental phases of 5 classes; 3) 40 classes for base phase and 20 incremental phases of 3 classes. *For TinyImageNet*, we start by training the model with 100 classes in the base phase and distribute the remaining classes into three incremental settings: 1) 5 incremental phases of 20 classes; 2) 10 incremental phases of 100 classes; 3) 20 incremental phases of 5 classes.

**Implementation details.** Our method is implemented with PyCIL [65]. For a fair comparison with [6], we adopt ResNet-18 [66] as the backbone network. The batch size is set to 64 for CIFAR-100 and TinyImageNet and 128 for ImageNet-Subset and ImageNet-1K. During training, the model is optimized by the Adam optimizer with $\beta_1 = 0.9$, $\beta_2 = 0.999$ and $\epsilon = 1e^{-8}$ (weight decay 2e-4). For ImageNet-1K, the learning rate starts at 0.0005 for all phases. The learning rate decays to 1/10 of the previous value every 70 epochs (160 epochs in total) in the base phase and every 45 epochs (100 epochs in total) in each incremental phase. For other datasets, the learning rate starts from 0.001 and decays to 1/10 of the previous value every 45 epochs (100 epochs in total) for all phases. We use $\lambda = 0.5$ and $\gamma = 0.1$ for all datasets. Regarding the loss weights, for comprehensive performance considerations and with reference to previous studies [6; 51], we set $\alpha_1 = 10$, $\alpha_2 = 10$, and $\alpha_3 = 2$ for training. We conduct our experiments on an RTX4090 GPU.

**Metric.** We evaluate the methods in terms of average incremental accuracy. Average incremental accuracy $A_T$ is computed as the average of the accuracy of all phases (including the base phase) and is a fair metric to compare the overall incremental performance of different methods:

$$A_T = \frac{1}{T+1} \sum_{t=0}^{T} a_t, \tag{15}$$

where $a_t$ is the average accuracy over all seen classes on phase $t$.

## 4.2 Comparison with SOTA

We compare our method with the state-of-the-art (SOTA) methods of NECIL (EWC [23], LwF_MC [67], MUC [68], SDC [8], PASS [6], SSRE [7], SOPE [11], POLO [51], PRAKA [10] and NAPA-VQ [52]). *"Fine-tuning"* refers to continuously fine-tuning the network on the new task with only cross-entropy loss. *"Joint"* means that when learning a new task, all data from past tasks are available to jointly train the model, which can be considered as an upper bound of the CIL model. The results reported for PASS are obtained with self-supervised learning.

The quantitative comparisons of average incremental accuracy are reported in Tab. 1. In comparison with the SOTA, our method improves by 1.4% and 6.0% on CIFAR-100 and TinyImageNet datasets, respectively. To further investigate the behavior of different methods on larger data, we also evaluated their performance on ImageNet-Subset. Compared with suboptimal results, PRL achieves an average improvement of 3.6%. The outstanding performance on ImageNet-Subset demonstrates the reliability of our method. To provide a more nuanced view of the changes in performance of the different methods over the course of incremental learning, we show accuracy curves for CIFAR-100, TinyImageNet and ImageNet-Subset in Fig. 3. The accuracy of our method remains ahead as we continue to learn new tasks. By prospective learning, our approach demonstrates strengths early on that will be maintained and even enlarged over the course of continuously learning new tasks.

Table 1: Quantitative comparisons of the average incremental accuracy (%) with other methods on CIFAR-100, TinyImageNet and ImageNet-Subset. *P* represents the number of incremental phases. The best performance is shown in **bold**, and the sub-optimal performance is underlined. The relative improvement compared to the SOTA NECIL methods is shown in red.

| Methods | CIFAR-100 | | | TinyImageNet | | | ImageNet-Subset | | |
|---|---|---|---|---|---|---|---|---|---|
| | *P*=5 | *P*=10 | *P*=20 | *P*=5 | *P*=10 | *P*=20 | *P*=5 | *P*=10 | *P*=20 |
| Fine-tuning | 23.15 | 12.96 | 7.93 | 18.64 | 10.68 | 5.75 | 23.43 | 13.12 | 7.96 |
| Joint | 76.72 | 76.72 | 76.72 | 63.08 | 63.08 | 63.08 | 78.94 | 78.94 | 78.94 |
| EWC [23] | 24.48 | 21.20 | 15.89 | 18.80 | 15.77 | 12.39 | — | 20.40 | — |
| LwF_MC [67] | 45.93 | 27.43 | 20.07 | 29.12 | 23.10 | 17.43 | — | 31.18 | — |
| MUC [68] | 49.42 | 30.19 | 21.27 | 32.58 | 26.61 | 21.95 | — | 35.07 | — |
| SDC [8] | 56.77 | 57.00 | 58.90 | — | — | — | — | 61.12 | — |
| PASS [6] | 63.47 | 61.84 | 58.09 | 49.55 | 47.29 | 42.07 | 64.40 | 61.80 | 51.29 |
| SSRE [7] | 65.88 | 65.04 | 61.70 | 50.39 | 48.93 | 48.17 | — | 67.69 | — |
| SOPE [11] | 66.64 | 65.84 | 61.83 | 53.69 | 52.88 | 51.94 | — | 69.22 | — |
| POLO [51] | 68.95 | 68.02 | 65.71 | 54.90 | 53.38 | 49.93 | 70.81 | 69.11 | — |
| PRAKA [10] | 70.02 | 68.86 | 65.86 | 53.32 | 52.61 | 49.83 | 69.81 | 68.98 | 63.95 |
| NAPA [52] | 70.44 | 69.04 | 67.42 | 52.77 | 51.78 | 49.51 | 69.15 | 68.83 | 63.09 |
| PRL (Ours) | **71.26** | **70.17** | **68.44** | **58.12** | **57.24** | **54.51** | **72.85** | **71.54** | **66.88** |
| Improvement | +0.82 | +1.13 | +1.02 | +3.22 | +3.86 | +2.57 | +2.04 | +2.32 | +2.93 |

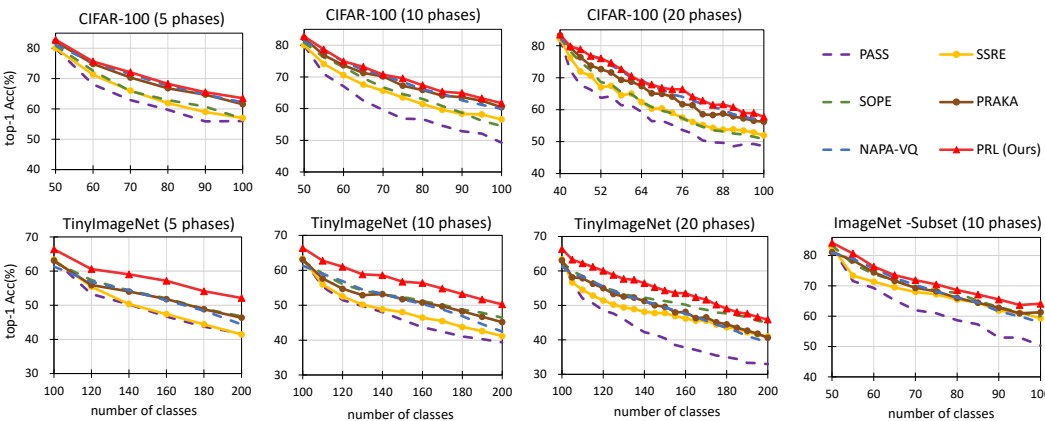

Figure 3: Detailed accuracy curves showing the top-1 accuracy of each incremental phase on CIFAR-100, TinyImageNet and ImageNet-Subset.

## 4.3 Ablation Study

To analyze the impact of each component in our method, we perform several ablation studies on CIFAR-100 and TinyImageNet datasets. We use the prototype augmentation technique in [10] as eq. (5) in our baseline. As shown in Tab. 2, The first line shows the performance of our baseline model. Our baseline is strong due to using the prototype augmentation in [10]. Even on the strong baseline model, both preemptive embedding squeezing (PES) constraint and prototype-guided representation update (PGRU) strategy can bring considerable performance improvements. Furthermore, the table shows that PES plays a more central role than PGRU. This is reasonable since the space reserved by PES for future classes is the basis for the PGRU to guide new classes to embed in the representation space during the incremental phase.

## 4.4 Analysis

**Visualization.** To analyze the impact of PRL on representation learning, we visualize the embedding space of 2D feature vectors on CIFAR-100 (5 phases) with t-SNE [69] in Fig. 4. Specifically, we (1)

Table 2: Ablation study (in average incremental accuracy) of our method on CIFAR-100 and TinyImageNet datasets.

| Methods | CIFAR-100 | | | TinyImageNet | | |
|---|---|---|---|---|---|---|
| | $P$=5 | $P$=10 | $P$=20 | $P$=5 | $P$=10 | $P$=20 |
| baseline | 69.25 | 68.52 | 65.93 | 55.04 | 54.15 | 51.65 |
| baseline w/ PES | 70.57 | 69.64 | 67.58 | 57.08 | 55.84 | 53.58 |
| baseline w/ PGRU | 70.36 | 69.23 | 67.17 | 56.79 | 56.05 | 53.16 |
| PRL | **71.26** | **70.17** | **68.44** | **58.12** | **57.24** | **54.51** |

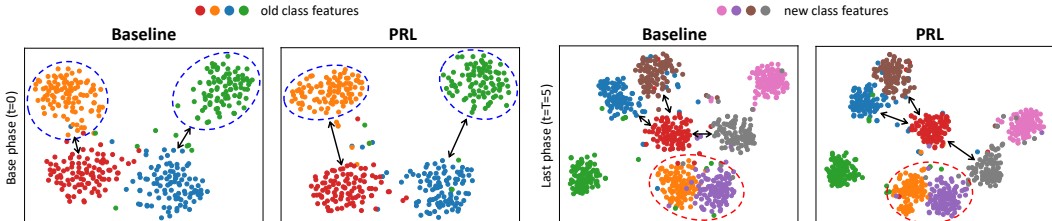

Figure 4: Visualization of the impact of PRL on the feature representations. Dashed circles and arrows highlight observable differences between baseline and PRL. PRL visually concentrates the distribution of features within classes, disperses the distribution of features between classes, and mitigates inter-class confusion.

visualize the features of a randomly selected subset of classes from $D_0$ (old class features) after the base phase, and (2) visualize the old class features along with a subset of classes from $D_T$ (new class features) after the last phase. As shown in the first row, once the training of the base phase ($t = 0$) is complete, the model integrated with PRL has more tightly clustered intra-class distributions (blue circles) and more dispersed inter-class distributions (↔). Thus, more space is reserved for learning new classes. The second row is visualized after the last phase ($t = T = 5$). It can be observed that the overlap (red circles) in the baseline model increases, causing confusion between the old and new classes. In contrast, PRL reduces the overlap between classes, making them easier to distinguish. Moreover, the new classes are farther away from the old ones (↔) compared to the baseline.

**Comparison of the confusion matrix.** Figure 5 compares the confusion matrices obtained by fine-tuning, PASS [6], NAPA-VQ [52] and our PRL on CIFAR-100. The diagonal entries indicate correct classification, while the non-diagonal entries indicate misclassification. Due to the forgetting of old classes, fine-tuning produces predictions that are biased toward the most recent classes, showing a strong confusion on the last task. PASS clearly mitigates this confusion but still predicts more intensively on recent tasks. The predictions of NAPA-VQ are largely centered on the diagonal, but its predictions are more accurate for the initial classes that appear in the base phase (the red patches are more localized in the first half of the diagonal). In contrast, there are more red patches visible along the diagonal and more evenly distributed in the confusion matrix of PRL, which explains the higher average accuracy of our method compared to NAPA-VQ and the absence of a serious bias towards either new or old classes.

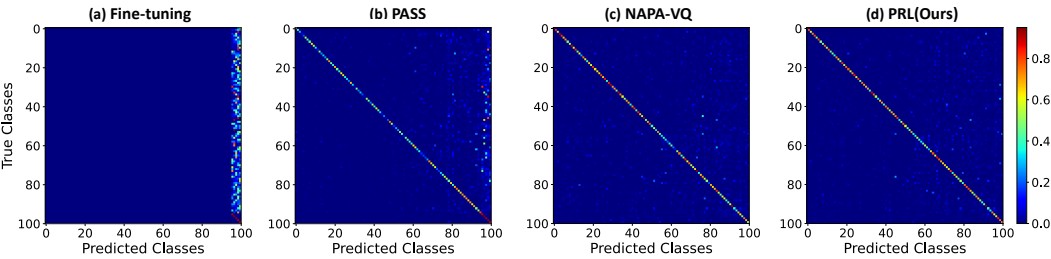

Figure 5: The comparison of confusion matrix of fine-tuning, PASS, NAPA-VQ and our method on CIFAR-100 (10 phases).

**Plasticity and stability analysis.** An incremental learner should acquire new knowledge of the current task for the sake of plasticity and also preserve knowledge from previous tasks for the sake of stability [70; 71]. We present an analysis of the plasticity and stability of the different methods in Fig. 6. First, we observe a gradual decline in average performance on past tasks during incremental learning. This is rational because experiencing more tasks also results in heavier catastrophic forgetting. Nonetheless, our method exhibits better stability due to less degradation and consistently superior average performance on old tasks. Then we turn our attention to the current task and also found a performance degradation

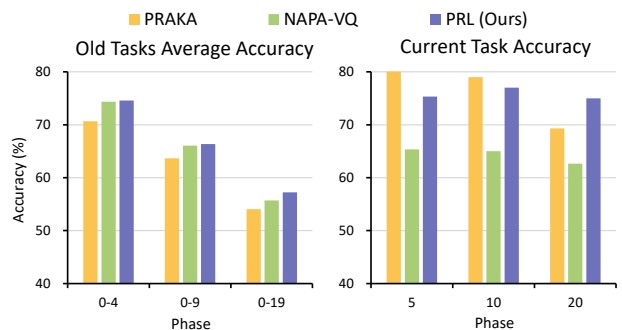

Figure 6: During incremental learning, our method shows less performance degradation on past tasks. Meanwhile, in contrast to other methods other methods whose performance on new tasks declines as the number of tasks increases or remains poor, our method shows good plasticity in the performance of new tasks.

as more and more tasks are learned. This corresponds to a gradual reduction in plasticity since tasks are sampled uniformly from the set of possible tasks, which is consistent with observations from previous studies [72; 73]. PRAKA [10] starts with good performance, but its plasticity degrades as more tasks are learned. NAPA-VQ consistently performs poorly on the current task, which is also in line with the results in Fig. 5. Remarkably, PRL maintains a good performance on the current task and has yet to show a visible decline. In general, our method achieves a better trade-off between stability and plasticity.

## 5   Conclusion and Limitation

In this work, we consider the conflict between old and new classes in NECIL from a prospective view. In the base phase, we construct a preemptive embedding squeezing constraint to reserve space for future classes by enforcing intra-class concentration and inter-class reserved separation. In the incremental phase, we propose a prototype-guided representation update (PGRU) strategy, which reduces the impact on the old class during model update by keeping the new class embedding away from the old class prototype. In cases where exemplars cannot be saved, waiting until the conflict arrives could exacerbate the problem, and we offer a novel solution. Through extensive experiments on four public benchmarks, our method exhibits excellent average performance and can provide a good balance between stability and plasticity. However, since the number and distribution of unknown classes cannot be predicted, how to rationally allocate the space of base classes in prospective learning is open to further discussion.

## Acknowledgments

This work is partially supported by National Natural Science Foundation of China under Grant (62176188, 62361166629, 62225113) and Key Research and Development Project of Hubei Province (2022BAD175). The numerical calculations in this paper have been done on the supercomputing system in the Supercomputing Center of Wuhan University.

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

# A Appendix / supplemental material

## A.1 Detailed Description of the Accuracy Curve

To facilitate comparison of future work with our method, we provide detailed values of the accuracy curves in Tab. 3, Tab. 4 and Tab. 5, where 'A' represents the CIFAR-100 dataset, 'B' represents the TinyImageNet dataset and 'C' represents the ImageNet-Subset dataset, respectively.

Table 3: Detailed values of accuracy under the setting of 5 phases.

| Dataset | Phase | | | | | |
|---|---|---|---|---|---|---|
| | 0 | 1 | 2 | 3 | 4 | 5 |
| A | 82.80 | 75.65 | 72.10 | 68.26 | 65.52 | 63.44 |
| B | 66.58 | 60.58 | 59.04 | 57.14 | 54.10 | 52.13 |
| C | 84.52 | 77.90 | 72.32 | 69.72 | 67.16 | 65.44 |

Table 4: Detailed values of accuracy under the setting of 10 phases.

| Datasets | Phase | | | | | | | | | | |
|---|---|---|---|---|---|---|---|---|---|---|---|
| | 0 | 1 | 2 | 3 | 4 | 5 | 6 | 7 | 8 | 9 | 10 |
| A | 82.80 | 78.76 | 74.90 | 73.18 | 70.71 | 69.53 | 67.35 | 65.36 | 64.90 | 63.24 | 61.71 |
| B | 66.58 | 62.75 | 61.02 | 58.83 | 58.57 | 56.73 | 56.34 | 54.79 | 53.18 | 51.64 | 50.25 |
| C | 84.52 | 80.69 | 76.37 | 73.57 | 71.89 | 70.51 | 68.6 | 67.13 | 65.53 | 63.68 | 64.10 |

Table 5: Detailed values of accuracy under the setting of 20 phases.

| Datasets | Phase | | | | | | | | | |
|---|---|---|---|---|---|---|---|---|---|---|
| | 0 | 1 | 2 | 3 | 4 | 5 | 6 | 7 | 8 | 9 |
| A | 83.45 | 79.81 | 78.85 | 76.80 | 76.06 | 74.64 | 72.64 | 70.52 | 68.27 | 67.84 |
| B | 66.38 | 63.26 | 62.22 | 61.19 | 60.07 | 58.85 | 57.68 | 57.46 | 56.45 | 55.31 |
| C | 84.75 | 80.84 | 77.91 | 78.08 | 75.27 | 74.55 | 73.31 | 69.38 | 67.75 | 65.85 |

| Datasets | Phase | | | | | | | | | | |
|---|---|---|---|---|---|---|---|---|---|---|---|
| | 10 | 11 | 12 | 13 | 14 | 15 | 16 | 17 | 18 | 19 | 20 |
| A | 66.80 | 66.36 | 66.37 | 64.06 | 62.84 | 61.46 | 61.61 | 60.77 | 58.98 | 58.94 | 57.74 |
| B | 54.34 | 53.61 | 53.51 | 52.41 | 51.69 | 50.23 | 49.06 | 48.05 | 47.58 | 47.63 | 45.58 |
| C | 63.66 | 62.36 | 62.18 | 60.86 | 60.22 | 60.85 | 57.73 | 57.91 | 57.30 | 57.32 | 56.52 |

**Evaluation on Large Datasets and Robustness.** To further demonstrate the effectiveness of our method, we evaluated it on a large-scale dataset — ImageNet-1K. *For ImageNet-1K, we allocate 500 classes for the base phase and 50 classes for each of the 10 incremental phases.* As shown in Table 6, our method shows an improvement of 1.9% compared to the suboptimal results. The results of our method are obtained by averaging three replicate experiments, and we set a different random seed for each run. To illustrate the stability of our method, we report the standard deviation of these three results. As shown in Tab. 6, the random seed has little impact on the results of our approach.

Table 6: The number after $\pm$ in the last line represents the standard deviation of three different runs.

| Methods | CIFAR-100 | | | TinyImageNet | | |
|---|---|---|---|---|---|---|
| | *P*=5 | *P*=10 | *P*=20 | *P*=5 | *P*=10 | *P*=20 |
| SOPE [11] | 66.64 | 65.84 | 61.83 | 53.69 | 52.88 | 51.94 |
| POLO [51] | 68.95 | 68.02 | 65.71 | 54.90 | 53.38 | 49.93 |
| NAPA [52] | 70.44 | 69.04 | 67.42 | 52.77 | 51.78 | 49.51 |
| PRL (Ours) | **71.26**±0.19 | **70.17**±0.31 | **68.44**±0.24 | **58.12**±0.48 | **57.24**±0.41 | **54.51**±0.36 |

| Methods | ImageNet-Subset | | | ImageNet-1K |
|---|---|---|---|---|
| | *P*=5 | *P*=10 | *P*=20 | *P*=10 |
| SOPE [11] | — | 69.22 | — | 60.20 |
| POLO [51] | 70.81 | 69.11 | — | 61.53 |
| NAPA [52] | 69.15 | 68.83 | 63.09 | 54.21 |
| PRL (Ours) | **72.85**±0.25 | **71.54**±0.27 | **66.88**±0.37 | **62.74**±0.34 |

Table 7: We report the performance gain of average incremental accuracy by applying PRL to other NECIL baselines. Absolute gains are marked in (red).

| Methods | CIFAR-100 | | |
|---|---|---|---|
| | *P*=5 | *P*=10 | *P*=20 |
| IL2A [38] | 67.35 | 61.03 | 60.67 |
| +PRL | 69.53 (+2.18) | 62.49 (+1.46) | 62.36(+1.69) |
| PASS [6] | 63.47 | 61.84 | 58.09 |
| +PRL | 66.22 (+2.75) | 62.85 (+1.01) | 58.85 (+0.76) |

**Plug-and-play with other NECIL methods.** Existing NECIL methods mainly focus on backward-looking means of resolving conflicts between old and new classes, which does not contradict our prospective learning. Therefore, we integrate PRL into the existing NECIL methods. Tab. 7 illustrates the performance gains achieved by incorporating PRL in these methods. In the setting of the CIFAR dataset with three different lengths of task sequences, PRL improved accuracy by an average of 2.7% for IL2A [38] and 2.3% for PASS [6], which demonstrates the good compatibility of our method.

### A.2 Impact of the hyper-parameter

To investigate the sensitivity of our method to the hyper-parameters $\lambda$ and $\gamma$, we performed ablation experiments on three settings (5 phases, 10 phases and 20 phases) of the CIFAR-100 dataset. In Fig. 7 we show the impact of $\lambda$, which controls the priority ratio of intra-class constraints and inter-class constraints. A smaller $\lambda$ means that the preemptive embedding squeezing (PES) is more concerned with intra-class concentration. Conversely, for a larger $\lambda$, more emphasis is placed on inter-class separation. When the value of $\lambda$ is either too large or too small, the performance of our method degrades, indicating that there is a need to maintain a certain balance between the intra-class constraint and the inter-class constraint. The best performance is achieved when $\lambda$ is equal to 0.5, suggesting that for prospective learning in NECIL, intra-class concentration could be more important than inter-class separation.

We also provide an analysis of the impact of the hyper-parameters $\gamma$ in Fig. 8. The performance of our method is stable on the 5 phase setting. The performance of the model gradually increases as $\gamma$ increases on the 10-phase and 20-phase settings, peaking at $\gamma = 0.1$. However, continued increase in the values of $\gamma$ leads to a decline in model performance. We argue that too large loss weights cause PES to interfere with the optimization of cross-entropy for classification performance. In addition, our method is more sensitive to the values of $\lambda$ and $\gamma$ when there are more tasks (20 phases) to learn.

For the hyperparameter of loss weight, we set $\alpha_1 = 10$, $\alpha_2 = 10$, and $\alpha_3 = 2$ by default. When a sensitivity analysis is performed on one of the hyperparameters, default settings are used for the remaining hyperparameters. As shown in Fig. 9, the left column shows the effect of changing the value of each hyperparameter on the average incremental accuracy of our method, and the right

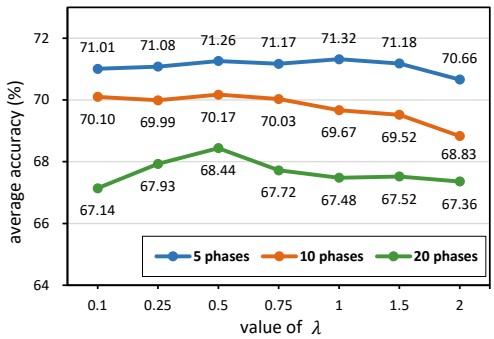

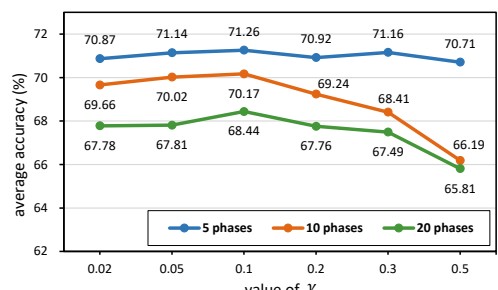

Figure 7: Impact of the hyper-parameter $\lambda$ in our preemptive embedding squeezing, which controls the priority ratio of intra-class constraints and inter-class constraints. Larger values of $\lambda$ represent a stronger inter-class separation.

Figure 8: Impact of the hyper-parameter $\gamma$, which controls for the weight of the PES loss. Larger values of $\lambda$ represent that the PES loss exerts a greater influence in the base phase of training compared to the cross-entropy loss.

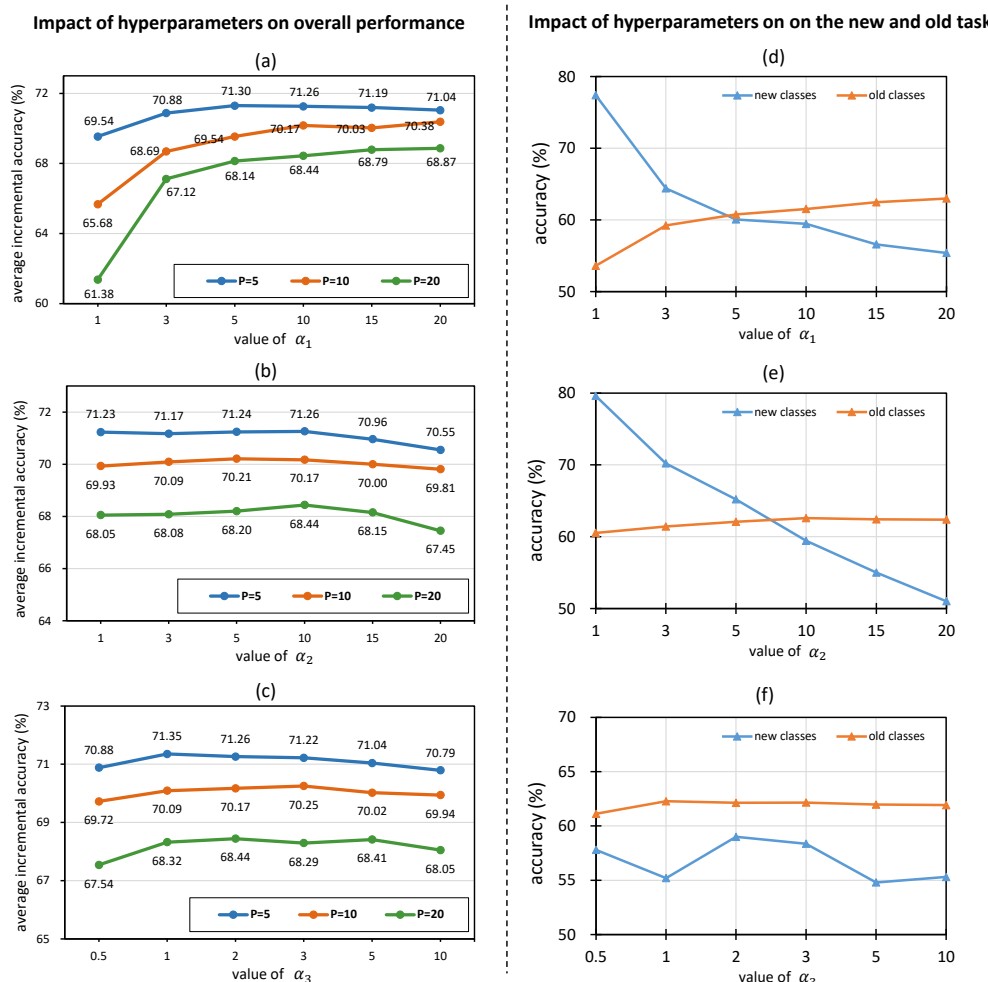

Figure 9: The figures in the **left** column show the effect of changing the value of each hyperparameter on the average incremental accuracy of our method on CIFAR-100 dataset, where 'P' denotes the number of incremental phases. The figures in the **right** column show the effect of changing the value of each hyperparameter in the last phase on the accuracy on the new and old tasks on CIFAR-100 dataset (P=10), respectively.

column shows the effect of changing the value of each hyperparameter in the last phase on the accuracy on the new and old tasks, respectively.

Among the three hyperparameters in eq. (14), $\alpha_1$ and $\alpha_2$ are common in previous NECIL methods and represent the weights of distillation loss and prototype loss, respectively. The main role of these two loss functions is to maintain the pre-existing knowledge of the model. Therefore, as shown in Fig. 9 (d) and Fig. 9 (e), as $\alpha_1$ and $\alpha_2$ get larger, the optimization of the model will be biased towards maintaining stability at the expense of plasticity, resulting in the model performing better on the old task and worse on the new task. It can be seen that as the value of $\alpha_1$ and $\alpha_2$ increases to a certain level its performance improvement on old tasks slows down. Excessively large values of and will bring much less gain on the old task than they will hurt performance on the new task. For the consideration of comprehensive performance and with reference to previous works [6; 51], we set $\alpha_1 = 10$ and $\alpha_2 = 10$ for our method.

Then $\alpha_3$ controls the loss of the Prototype-Guided Representation Update (PGRU) proposed in this paper. In Fig. 9 (c), as $\alpha_3$ increases PGRU comes into play. The effect of increasing $\alpha_3$ on the overall performance of the algorithm fluctuates, which may be caused by overly strict constraints on the learning of new class representations. Overall, our algorithm is robust to the hyperparameters.

