# OpenReview forum: "Prospective Representation Learning for Non-Exemplar Class-Incremental Learning"
_NeurIPS.cc/2024/Conference — NeurIPS 2024 poster_

### Official Review · Reviewer_nQN3 · 2024-07-04

**Soundness:** 3
**Presentation:** 3
**Contribution:** 2
**Rating:** 6
**Confidence:** 4

**Summary:**

This paper aims to solve the catastrophic forgetting in non-exemplar class-incremental learning. The authors propose Prospective Representation Learning (PRL) to prepare representation space for classes in later tasks in advance. Such forward compatible method first squeezes the embedding distribution of the current classes to reserve space for forward compatibility with future classes, then makes the new class features away from the saved prototypes of old classes. Extensive experiments are performed to demonstrate that the method is effective.

**Strengths:**

1. The paper is easy to understand and follow.
2. The reported performance of the proposed method seems good, especially in TinyImageNet.
3. The illustration of the method is clear.

**Weaknesses:**

1. The novelty of the method could be one of the concerns. There are previous works considering forward-compatible incremental learning [1,2]. The embedding space reservation is not a new concept, which has been proposed in [2]. This paper lacks related works and discussion on the advantages over other forward compatible methods and comparison with them.
2. Lack of evaluation on large datasets like Imagenet1000.

[1] Shi et al. Mimicking the Oracle: An Initial Phase Decorrelation Approach for Class Incremental Learning.

[2] Zhou et al. Forward Compatible Few-shot Class-Incremental Learning.

**Questions:**

1. What are the advantages of PRL over other forward-compatible methods?
2. Can you provide the evaluation results on large datasets?

**Limitations:**

The paper has a section of limitation.

---

> ### Author Rebuttal · Authors · 2024-08-04
>
> Thank you for your constructive feedback! We hope the following responses can address your concerns.
>
> ---
>
> **W1 & Q1: What are the advantages of PRL over other forward-compatible methods?**
>
> **A1:** First, unlike previous works, PRL targets CIL in exemplar-free scenarios (NECIL). NECIL requires the algorithm to learn a unified model without access to any data from previous tasks, which means that the commonly used memory buffers are not available thus more challenging.
>
> Second, compared to previous works, PRL considers how to resolve conflicts between new and old classes during the incremental phases, in addition to reserving space in the initial phase. Zhou et al [2] also mentioned that such conflicts need to be taken into account when it transforms into a CIL problem (*i.e.*, there are enough instances of the new classes). Compared to the work of Zhou et al [2], which presupposes unseen classes by mixing instances from the initial task, our approach improves the forward compatibility of the model without interfering with the learning of the initial task itself.
>
> The reference works you provided are inspiring to our research. We will add a discussion of the works on forward-compatible incremental learning in the final version of our paper.
>
> ---
>
> **W2 & Q2: Lack of evaluation on large datasets like Imagenet1000.**
>
> **A2:** The following table compares the average incremental accuracies of the different methods for the 10 phases setting on ImageNet-1k. We will provide more experimental results on ImageNet-1k in the final version of the paper.
>
> |  | &ensp; PASS &ensp; | &ensp; SSRE &ensp; |  &ensp; SOPE &ensp; | &ensp; POLO &ensp; | &ensp; NAPA-VQ &ensp; | &ensp; PRL (Ours) &ensp; |
> | --- | :---: | :---: | :---: | :---: | :---: | :---: |
> | Accuracy (%) | 55.90 | 58.12 | 60.20 | 61.53 | 54.21 | **62.74** |
>
> We are looking forward to answering any follow up questions during the discussion period.

---

> > ### Comment · Reviewer_nQN3 · 2024-08-12
> > **Thank You for the response**
> >
> > The authors' response resolves most of my concerns, I decide to raise my score.

---

> > > ### Author Response · Authors · 2024-08-12
> > > **Official Comment by Authors**
> > >
> > > Thank you for taking the time to read our response and increasing your score! We are glad to hear that our response addressed your concern.

---

### Official Review · Reviewer_SYbV · 2024-07-09

**Soundness:** 3
**Presentation:** 2
**Contribution:** 2
**Rating:** 4
**Confidence:** 3

**Summary:**

This work introduces a Prospective Representation Learning (PRL) scheme to prepare the model for handling conflicts of the balance between the old and new classes in Non-exemplar class-incremental learning (NECIL). The author proposes to squeeze the embedding distribution of the current classes in the base phase and make the new class features away from the saved prototypes of old classes in the incremental phase, improving the balance of old and new classes in the existing NECIL baselines.

**Strengths:**

1. This work proposes the Prospective Representation Learning (PRL) scheme to solve the balance problem between old and new classes in NECIL tasks.
2. The proposed PRL scheme is plug and play, making it convenient to combine with other models.

**Weaknesses:**

1.  Few analysis was conducted on the loss weight α in formula 14.
2.  There are some grammar or writing errors in the text, such as the repeated appearance of the "propose" on line 111.

**Questions:**

1. Why not perform Preemptive Embedding Squeezing (PES) on new classes during the incremental phase? Will this not cause overlap in the feature space of the new classes?
2. What does the $L_{IIC}$ of Formula 9 indicate? I do not find the corresponding explanation. (ps: I guess the author may have mistakenly wrote $L_{PES}$ as $L_{IIC}$)
3. What do the old class prototypes of formula 11 and the potential spatial features of formula 12 specifically refer to, and is there a relationship between them?

---

> ### Author Rebuttal · Authors · 2024-08-06
>
> Thank you so much for the insightful questions! We will revise our manuscript accordingly and address your questions below.
>
> ---
> **W1: Few analysis was conducted on the loss weight α in formula 14.**
>
> **A1:** We set $\alpha_1=10$, $\alpha_2=10$ , and $\alpha_3=2$ by default. Due to rebuttal limitations, the experimental results are shown in the **pdf file** attached to the Author Rebuttal. In Eq.14, $\alpha_1$ and $\alpha_2$ are common in previous NECIL methods and represent the weights of distillation loss and prototype loss, respectively. The main role of these two loss functions is to maintain the pre-existing knowledge of the model. Therefore, as shown in Fig.(d) and Fig.(e), as $\alpha_1$ and $\alpha_2$ get larger, the optimization of the model will be biased towards maintaining stability, resulting in the model performing better on the old task and worse on the new task. For the consideration of comprehensive performance, we set $\alpha_1 = 10$ and $\alpha_2=10$ for PRL.
>
> Then $\alpha_3$ controls the loss of the Prototype-Guided Representation Update (PGRU) proposed in this paper. In Figure (c), as $\alpha_3$ increases PGRU comes into play. The effect of growing $\alpha_3$ on the overall performance fluctuates, which may be caused by overly strict constraints on the learning of new class representations. Overall, our algorithm is relatively robust to the choice of hyperparameters.
>
> ---
> **W2 & Q2: Some grammar or writing errors.**
>
> **A2:** Thank you for pointing out our error. As you guessed, $\mathcal{L}_ {IIC}$ was a writing error, and should actually be $\mathcal{L}_ {PES}$. We will carefully scrutinize the main text and the appendix, and ensure such errors are eliminated in the final version of the paper.
>
> ---
> **Q1: Why not perform Preemptive Embedding Squeezing (PES) on new classes during the incremental phase? Will this not cause overlap in the feature space of the new classes?**
>
> **A3:** First, going into the incremental phase, the main problem faced by the model shifts from adjusting the relationship between new classes to dealing with the overlap between the new classes and the old ones, especially in the absence of samples from the old classes. We therefore propose Prototype-Guided Representation Update (PGRU) to alleviate this problem. Due to the shift of the main problem, the improvement from performing PES in the incremental phase is not obvious.
>
> We further reflect on this issue. This could be caused by the fact that as incremental learning goes on and on, the number of new classes is relatively small compared to the old ones. Therefore we try to increase the number of classes in the incremental phase for our experiments on CIFAR-100. In the following table, we performed only one incremental phase and set the number of classes $C$ in the incremental phase to 5, 10, 20, and 50, respectively. It can be seen that the effect of PES is gradually apparent when the number of new classes increases.
>
> | |C=5|C=10|C=20|C=50|
> |---|:---:|:---:|:---:|:---:|
> |w/ PES|80.97|79.10|77.78|73.96|
> |w/o PES|80.84|79.05|77.34|73.37|
>
> Certainly, the ratio of the number of new classes to the number of old classes naturally decreases as the tasks learned by the model accumulate, so we only employ PES in the initial phase. Previous work [1] that considered forward compatibility also optimized the learning of the model only in the initial phase.
>
> Second, in the incremental phase, due to the sufficient number of samples in the new classes, the cross-entropy loss also makes the features of the new classes distinguish from each other and reduce the overlap of the new classes.
>
> ---
> **Q3: What do the old class prototypes of formula 11 and the potential spatial features of formula 12 specifically refer to, and is there a relationship between them?**
>
> **A4:** We have described the old class prototypes and the features in the potential space in Sections 3.2 (line154-158) and Section 3.3 (line195-200), respectively. At the end of each phase, a prototype is computed and saved for each class of the current task. As shown in Eq.4, the prototype is usually the average of all the features of the class. This is widely used in existing NECIL methods [2, 3]. The potential spatial features refer to features $\mathcal{P}_ {\phi_ t}(\mathcal{F}_ {\theta_ {t-1}}(x^i))$ that have been projected, where $x^i$ denotes the sample of the current class, $\mathcal{F}_ {\theta_ {t-1}}(x^i)$ is the feature extracted by the teacher model (the model trained from the previous phase).
>
> Old class prototypes cannot be updated after saving due to the absence of old class data. Since the teacher model $\mathcal{F}_ {\theta_ {t-1}}$ is frozen, $\mathcal{F}_ {\theta_ {t-1}}(x^i)$ also does not change in the current phase of training. Therefore, we introduce a projector $\mathcal{P}_ {\phi_t}$ to project the prototypes $\boldsymbol{p}^c$ and $\mathcal{F}_ {\theta_ {t-1}}(x^i)$ into a potential space. Optimizing the projector via Eq.11 can change the relationship between $\boldsymbol{p}^c$ and $\mathcal{F}_ {\theta_ {t-1}}(x^i)$ in the potential space thus avoiding the new class features to be too close to the region where the old class prototype is located. Eq.12 is to align the features $\mathcal{F}_ {\theta_ {t}}(x^i)$ extracted by the current updated model $\mathcal{F}_ {\theta_ {t}}$ with the potential spatial features $\mathcal{P}_ {\phi_ t}(\mathcal{F}_ {\theta_ {t-1}}(x^i))$ that have been adjusted by the projector.
>
> We will optimize the expression to make this clearer in the final version.
>
> ---
> **Reference:**
>
> [1] Shi et al. Mimicking the Oracle: An Initial Phase Decorrelation Approach for Class Incremental Learning.
>
> [2] Zhu et al. Prototype augmentation and self-supervision for incremental learning.
>
> [3] Shi et al. Prototype reminiscence and augmented asymmetric knowledge aggregation for non-exemplar class-incremental learning.
>
> ---
> We are looking forward to answering any follow up questions during the discussion period.

---

> > ### Author Response · Authors · 2024-08-08
> > **Official Comment by Authors**
> >
> > Dear reviewer SYbV,
> >
> > We thank you sincerely for your time and effort in reviewing our manuscript and providing valuable suggestions. We have provided detailed responses to your questions and hope that they adequately address your concerns. If you need further clarification or have any other questions, please feel free to discuss them with us! We are more than willing to continue our communication with you.
> > We would greatly appreciate it if you would update the rating by synthesizing other reviewers' comments as well as our responses.

---

> > > ### Comment · Reviewer_SYbV · 2024-08-11
> > > **Official Comment by Reviewer SYbV**
> > >
> > > Thanks to the authors for their detailed responses, but I still have some concerns about the PES method.
> > >
> > > Using only the CE loss can also achieve good performance as the authors mentioned, and based on the results provided on the CIFAR100, the improvement is not significant.

---

> ### Author Response · Authors · 2024-08-11
> **Responses to Comment by Reviewer SYbV**
>
> Thank you for your response. We suppose that you may have misunderstood our response and our experiments in rebuttal.
>
> Our proposed PES aims to construct a better initial embedding space. Therefore, in our initial submitted version, **PES is only performed in the first phase**. We have done extensive experiments in our submitted version (Table 2 and Figure 3) to **demonstrate the effectiveness of this manner.**
>
> We also list the results in the following. Results demonstrate that PES can significantly improve the forward compatibility of model since it reserves space to prepare for future classes. For continual learning on incremental new classes, to make new classes embedded in the space previously reserved, PES was not included in the incremental phase in our submitted version.
>
> |     | CIFAR-100 (5 phases) | CIFAR-100 (10 phases) | CIFAR-100 (20 phases) | TinyImageNet (5 phases) | TinyImageNet (10 phases) | TinyImageNet (20 phases) |
> | --- | :---: | :---: | :---: | :---: | :---: | :---: |
> | baseline | 69.25 | 68.52 | 65.93 | 55.04 | 54.15 | 51.65 |
> | baseline w/ PES | 70.57 | 69.64 | 67.58 | 57.08 | 55.84 | 53.58 |
>
> In the first round of review, as you suggested, we add experiments that using PES in the incremental phase, shown in the rebuttal, which lead your misunderstandings. The results have verfied that **adding PES in the incremental phase indeed does not bring in further improvement**. It means that **our original design in submitted version is reasonable and effective**. We do not need PES in the incremental phase.
>
> We hope this response clarifies your misunderstandings.

---

### Official Review · Reviewer_FnFn · 2024-07-12

**Soundness:** 2
**Presentation:** 3
**Contribution:** 2
**Rating:** 5
**Confidence:** 5

**Summary:**

This paper focuses on non-exemplar class-incremental learning, specifically addressing the challenge of balancing old and new classes. The author introduces Prospective Representation Learning (PRL), which involves constructing a preemptive embedding squeezing constraint to allocate space for future classes. Additionally, the author proposes a prototype-guided representation update strategy.

**Strengths:**

1.The Preemptive Embedding Squeezing (PES) constrains the current class space to prepare for accommodating future new classes.

2.The Prototype-Guided Representation Update (PGRU) strategy ensures that features of new classes remain distinct from prototypes of old classes in the latent space.

3.The writing is clear.

4.The paper includes extensive experiments.

**Weaknesses:**

1.What is the meaning of IIC in equation 9? The paper does not explain its meaning; I guess it stands for PES.
2.Many previous works have studied mapping class centers to different subspaces (orthogonal). The paper should compare similar works to highlight the differences and advantages of the proposed method. As indicated by the following references: [1,2,3].
3.The PASS paper also uses prototype augmentation (and proposes other methods), but your baseline is higher than PASS, especially in TinyImageNet P=20, by almost 10%. The author should explain the advantages of using prototype augmentation in the baseline or provide experimental results without prototype augmentation.
4.In equation 14, there are many hyperparameters, /alpha1,2,3. The author should provide more sensitivity analysis of the hyperparameters to make the experiments more thorough.

[1] Chaudhry A, Khan N, Dokania P, et al. Continual learning in low-rank orthogonal subspaces[J]. Advances in Neural Information Processing Systems, 2020, 33: 9900-9911.
[2] Guo Y, Hu W, Zhao D, et al. Adaptive orthogonal projection for batch and online continual learning[C]//Proceedings of the AAAI Conference on Artificial Intelligence. 2022, 36(6): 6783-6791.
[3] French R M. Dynamically constraining connectionist networks to produce distributed, orthogonal representations to reduce catastrophic interference[C]//Proceedings of the sixteenth annual conference of the cognitive science society. Routledge, 2019: 335-340.

**Questions:**

1.Many previous works have studied mapping class centers to different subspaces (orthogonal). The paper should compare similar works to highlight the differences and advantages of the proposed method. As indicated by the following references: 1, 2.
2.The PASS paper also uses prototype augmentation (and proposes other methods), but your baseline is higher than PASS, especially in TinyImageNet P=20, by almost 10%. The author should explain the advantages of using prototype augmentation in the baseline or provide experimental results without prototype augmentation.

**Limitations:**

Yes

---

> ### Author Rebuttal · Authors · 2024-08-06
>
> Thank you for your thoughtful questions! We are glad that you found the paper easy to read and affirm our experiment. We hope that our response below will address your
> concerns.
>
> ---
>
> **W1: What is the meaning of IIC in equation 9? Is it a writing error?**
>
> **A1**: We are sorry for our writing error. It should be $\mathcal{L}_{PES}$ in Eq. 9. We will carefully check and ensure such errors are eliminated in the final version of the paper.
>
> ---
>
> **W2 & Q1: Discussion of the differences and advantages of the method proposed in this paper over previous studies on mapping class centers to different subspaces.**
>
> **A2:** First, unlike previous works, PRL targets CIL in exemplar-free scenarios (NECIL). PRL does not need to store exemplars of past tasks to promote subspace orthogonality.
>
> Second, previous works start to deal with the conflict between old and new tasks only when a new task arrives. However, due to the lack of samples from old tasks in NECIL, it is intractable to deal with this conflict using only the new task data. Instead, we consider reserving space for unknown new classes in the initial phase to prepare for conflict resolution in advance.
>
> The references you provided is enlightening to our research. We will include a discussion of the above methods in the Related Work section.
>
> ---
>
> **W3 & Q2: Explain the advantages of using prototype augmentation in the baseline compared to PASS.**
>
> **A3:** Since only one prototype is saved for each class, using one prototype to represent the old class distribution would lack diversity. Prototype augmentation helps to maintain the discrimination between old and new classes and prevents the decision boundary from being biased in favor of the new classes. From the perspective of our paper, both prototype augmentation and the knowledge distillation technique commonly used in NECIL are prompting the model to achieve backward compatibility, *i.e.*, to make the updated model compatible with classes that have already been learned. The PRL proposed in this paper, on the other hand, is prompting the model to achieve forward compatibility, i.e., enhancing the ability of the model to be compatible with unseen classes. Therefore, better performance can be achieved when both forward and backward compatibility of the model are improved.
>
> After PASS [1], many other prototype augmentation strategies have been proposed, such as [2, 3]. We adopt the prototype augmentation method called Prototype Reminiscence (PR) from [2] as our baseline, which is described in Section 3.2 (line 162).
>
> The experiments in the Appendix (Table 6) also demonstrate that our PRL can be combined with PASS or other methods in a plug-and-play manner and enhance its performance. We complement the performance of PASS combined with PRL on the **TinyImageNet** dataset in the following table, where 'P' denotes the number of incremental phases. It can be seen that the PRL brings a significant boost to PASS as well.
>
> |     | &ensp; P=5 &ensp;  | &ensp;  P=10 &ensp;  | &ensp;  P=20 &ensp; |
> | :---: | :---: | :---: | :---: |
> | PASS &ensp;| 49.55 | 47.29 | 42.07 |
> | PASS+PRL &ensp; | 52.19 | 50.38 | 42.63 |
>
> ---
>
> **W4: More sensitivity analysis of the hyperparameters.**
>
> **A4:** We set $\alpha_1=10$, $\alpha_2=10 $, and $\alpha_3=2$ by default. When a sensitivity analysis is performed on one of the hyperparameters, default settings are used for the remaining hyperparameters. Due to rebuttal limitations, the experiment results are shown in the **pdf file** attached to the Author Rebuttal. The figures in the left column show the effect of changing the value of each hyperparameter on the average incremental accuracy of our method. The figures in the right column show the effect of changing the value of each hyperparameter in the last phase on the accuracy on the new and old tasks, respectively.
>
> Among the three hyperparameters in Eq. 14, $\alpha_1$ and $\alpha_2$ are common in previous NECIL methods and represent the weights of distillation loss and prototype loss, respectively. The main role of these two loss functions is to maintain the pre-existing knowledge of the model. Therefore, as shown in Figure (d) and Figure (e), as $\alpha_1$ and $\alpha_2$ get larger, the optimization of the model will be biased towards maintaining stability at the expense of plasticity, resulting in the model performing better on the old task and worse on the new task. It can be seen that as the value of $\alpha_1$ and $\alpha_2$ increases to a certain level its performance improvement on old tasks slows down. Excessively large values of $\alpha_1$ and $\alpha_2$ will bring much less gain on the old task than they will hurt performance on the new task. For the consideration of comprehensive performance and with reference to previous works [1, 3], we set $\alpha_1 = 10$ and $\alpha_2=10$ for our method.
>
> Then $\alpha_3$ controls the loss of the Prototype-Guided Representation Update (PGRU) proposed in this paper. In Figure (c), as $\alpha_3$ increases PGRU comes into play. The effect of growing $\alpha_3$ on the overall performance of the algorithm fluctuates, which may be caused by overly strict constraints on the learning of new class representations. Overall, our algorithm is relatively robust to the choice of hyperparameters.
>
> We will add a clarification on hyperparameters in the final version.
>
> ---
>
> **Reference:**
>
> [1] Zhu F et al. Prototype augmentation and self-supervision for incremental learning. CVPR 2021.
>
> [2] Shi W et al. Prototype reminiscence and augmented asymmetric knowledge aggregation for non-exemplar class-incremental learning. ICCV 2023.
>
> [3] Wang S et al. Non-Exemplar Class-Incremental Learning via Adaptive Old Class Reconstruction. ACM MM 2023.
>
> ---
>
> We are looking forward to answering any follow up questions during the discussion period.

---

### Official Review · Reviewer_KX3q · 2024-07-14

**Soundness:** 3
**Presentation:** 3
**Contribution:** 3
**Rating:** 5
**Confidence:** 4

**Summary:**

The paper proposes a method to deal with incremental classification task in which no exemplars from the previously seen classes can be saved for usage during training on the newly arriving classes. The proposed method squeezes the embedding distribution of the current classes to reserve space for forward compatibility with future classes and reduces the impact of introducing new classes by trying to restrict the embeddings of the new classes in the regions not occupied by the previously seen classes. The method uses the class prototype of the previously seen classes but does not use any exemplars. The method uses Preemptive Embedding Squeezing and Prototype-Guided Representation Update to achieve the above goals.

**Strengths:**

The proposed method is based on the premise of compressing the feature space occupied by the previously seen classes to ensure less interference between old and new classes, which sounds logical.
The method seems to be clearly written.
The proposed method performs well on all the compared datasets.

**Weaknesses:**

The paper should include a separate section to discuss the difference between this method and other methods that employ feature space compression for incremental learning.
The standard deviation in the results between different runs is not mentioned.
Apart from accuracy improvement, does the proposed method involve fewer/more parameters as compared to the other compared methods? Is there any difference in training or testing time for the proposed method as compared to the others.

**Questions:**

The paper should include a separate section to discuss the difference between this method and other methods that employ feature space compression for incremental learning.
The standard deviation in the results between different runs is not mentioned.
Apart from accuracy improvement, does the proposed method involve fewer/more parameters as compared to the other compared methods? Is there any difference in training or testing time for the proposed method as compared to the others.

**Limitations:**

The paper mentions a limitation in that it is not able to rationally allocate the space of base classes since the number and distribution of unknown classes cannot be predicted. However, the authors did not experimentally demonstrate any such issue, which can be possibly done by changing the order of the classes between multiple runs.

---

> ### Author Rebuttal · Authors · 2024-08-05
>
> Thank you for your positive response! We are delighted that the reviewer has found our method clear and sound. We have addressed the main points and questions below.
>
> ---
>
> **Q1：Lack of discussion with other methods that employ feature space compression for incremental learning.**
>
> **A1**: Thanks to your suggestion, we will add a separate subsection to the Related Work section to discuss methods on feature space compression in incremental learning.
>
> ---
>
> **Q2: The standard deviation of the results between different runs that change the order of the classes.**
>
> **A2**: We perform three runs and use a different random seed to set the class order for each run. The table below shows the standard deviation of the three runs, where 'P' denotes the number of incremental phases and 'Tiny' denotes the TinyImageNet dataset. The experiments show that our method is robust to different class orders.
>
> |     | &nbsp; CIFAR100 (P=5) &nbsp; | &nbsp; CIFAR100 (P=10) &nbsp; | &nbsp; CIFAR100 (P=20) &nbsp; | &ensp; Tiny (P=5) &ensp; | &ensp; Tiny (P=10) &ensp; | &ensp; Tiny (P=20) &ensp; |
> | --- | :---: | :---: | :---: | :---: | :---: | :---: |
> | PRAKA | 68.95 | 69.02 | 65.71 | 54.90 | 53.38 | 49.93 |
> | NAPA-VQ | 70.44 | 69.04 | 67.42 | 52.77 | 51.78 | 49.51 |
> | PRL (Ours) &nbsp; | **71.26**$\pm$0.19 | **70.17**$\pm$0.31 | **68.44**$\pm$0.24 | **58.12**$\pm$0.48 | **57.24**$\pm$0.41 | **54.51**$\pm$0.36 |
>
> ---
>
> **Q3: Comparison on the number of model parameters, training time and testing time.**
>
> **A3**: The introduction of a projector in our method introduces additional parameters, but the number of parameters added is minimal compared to the network as a whole. And the forward propagation does not need to go through the projector during the test. The following table shows the model parameters of our method, the training time for one epoch in the last incremental phase and the testing time (on all classes) on **CIFAR-100**. The experimental results are performed with in the same environment.
>
> **Model parameters：**
>
> |     | &nbsp; PASS &nbsp; | &nbsp; PRAKA &nbsp; | &nbsp; NAPA-VQ &nbsp; | &nbsp; PRL (Ours) &nbsp; |
> | :---: | :---: | :---: | :---: | :---: |
> | Parameters &nbsp; | 10.93M | 11.02M | 11.18M | 11.30M |
>
> **Testing time:**
>
> |     | &nbsp; PASS &nbsp; | &nbsp; PRAKA &nbsp; | &nbsp; NAPA-VQ &nbsp; | &nbsp; PRL (Ours) &nbsp; |
> | :---: | :---: | :---: | :---: | :---: |
> | Test time &nbsp; | 1.82s | 2.38s | 4.31s | 1.74s |
>
> **Training time for one epoch:**
>
> |     | &emsp; P=5 &emsp; | &emsp; P=10 &emsp; | &emsp; P=20 &emsp; |
> | :---: | :---: | :---: | :---: |
> | PASS &nbsp; | 10.09s | 5.92s | 4.81s |
> | PRAKA &nbsp; | 7.62s | 4.27s | 3.57s |
> | NAPA-VQ &nbsp; | 20.06s | 10.62s | 6.54s |
> | PRL (Ours) &nbsp; | 7.97s | 4.54s | 3.55s |
>
> We are looking forward to answering any follow up questions during the discussion period.

---

### Author Rebuttal · Authors · 2024-08-07

We thank all reviewers for their positive view of our work and valuable feedback. We responded to reviewers' comments in individual replies to each reviewer with references to weakness (**W**) and questions (**Q**).

In response to the question of Reviewer FnFn and Reviewer SYbV about the analysis of the hyperparameters $\alpha$ in Eq. 14, we have attached a pdf file showing the experiment results. In the pdf file, the figures in the left column show the effect of changing the value of each hyperparameter on the average incremental accuracy of our method; the figures in the right column show the effect of changing the value of each hyperparameter in the last phase on the accuracy on the new and old tasks, respectively.

Please let us know if there are additional items or further clarifications/discussions we could address. We will incorporate clarifications and additions, as we specified in our replies, in the final version of our work.

---

### Comment · Area_Chair_eZ9W · 2024-08-09
**Discussion period instructions**

Dear Reviewers,

The authors have provided comprehensive rebuttals and tried to address the concerns raised in your reviews. Please take the time to review their responses carefully. If you have any further questions or require additional clarification, please engage in a discussion with the authors. Thank you for your continued efforts.

AC

---

### Comment · Area_Chair_eZ9W · 2024-08-13
**Please respond to the authors' rebuttal and comments if you haven't already done so**

Dear Reviewers,

As the discussion period is nearing its end, with less than one day remaining, I kindly urge you to respond to the authors' rebuttal and latest comments if you haven't already done so. Please don't hesitate to ask any remaining questions, as this will be our final opportunity to interact with the authors before the next phase of the review process.

Thank you very much for your dedication and effort.

Best regards,

AC

---

### Decision · Program_Chairs · 2024-09-25

**Decision:**

Accept (poster)

**Comment:**

This paper introduces a Prospective Representation Learning (PRL) scheme designed to manage conflicts between old and new classes in non-exemplar class-incremental learning. The proposed approach includes strategies for both the base and incremental stages, effectively reserving space in the learned representation and maintaining its structure as new classes are introduced. Experimental results demonstrate the effectiveness of PRL.

The paper has received mixed reviews, with one reviewer increasing their score after the rebuttal, while others did not respond to the updated results provided by the authors. The final scores after the rebuttal phase are (6, 5, 5, 4).

After a careful review, the AC finds that the paper does introduce a degree of novelty and the results validate its effectiveness. Given the significance of this approach to the field of class-incremental learning (CIL) and its potential to inspire future research, the AC suggests this paper as a borderline accept.

If the paper is accepted, the authors should carefully address the concerns raised by the reviewers, particularly the need for more detailed comparisons and in-depth analysis with baselines using CE losses, as well as further discussions of the related works mentioned by the reviewers.